# Different Original and Biosimilar TNF Inhibitors Similarly Reduce Joint Destruction in Rheumatoid Arthritis—A Network Meta-Analysis of 36 Randomized Controlled Trials

**DOI:** 10.3390/ijms20184350

**Published:** 2019-09-05

**Authors:** Niels Graudal, Benjamin Skov Kaas-Hansen, Louise Guski, Thorbjørn Hubeck-Graudal, Nicky J. Welton, Gesche Jürgens

**Affiliations:** 1The Lupus and Vasculitis Clinic VRR 4242, Copenhagen University Hospital, Blegdamsvej 9, DK 2100 Copenhagen, Denmark; 2Clinical Pharmacology Unit, Zealand University Hospital, Roskilde, Munkesøvej 18, 4000 Roskilde, Denmark; 3NNF Center for Protein Research, University of Copenhagen, Blegdamsvej 3B, 2200 Copenhagen N, Denmark; 4Department of Nuclear Medicine & PET-Centre, Aarhus University Hospital, Palle Juul-Jensens Boulevard 99, DK 8200 Aarhus, Denmark; 5Deparment of Population Health Science, Bristol Medical School, University of Bristol, Canynge Hall, 39 Whatley Rd, Bristol BS8 2PS, UK

**Keywords:** rheumatoid arthritis, tumor necrosis factor (TNF) inhibitors, joint destruction, randomized controlled trial, network meta-analysis

## Abstract

The effect of five approved tumour necrosis factor inhibitors (TNFi: infliximab, etanercept, adalimumab, certolizumab, and golimumab) on joint destruction in rheumatoid arthritis (RA) have been compared versus methotrexate (MTX) in randomized controlled trials (RCTs) but have not been compared directly to each other or to an otherwise untreated placebo control. The present analysis compares effects of standard doses, high doses, and low doses of TNFis on radiographic joint destruction in RA and relate these effects to MTX and placebo by means of a Bayesian network meta-analysis. We identified 31 RCTs of the effect of TNFis on joint destruction and 5 RCTs with controls, which indirectly could link otherwise untreated placebo controls to the TNFi treatments in the network. The previously untested comparison with placebo was performed to estimate not only the effect relative to another drug, but also the absolute attainable effect. Compared to placebo there was a highly significant inhibitory effect on joint destruction of infliximab, etanercept, adalimumab, certolizumab, and golimumab, which was about 0.9% per year as monotherapy and about 1.2% per year when combined with MTX. Although significantly better than MTX and placebo, golimumab seemed inferior to the remaining TNFis. There was no difference between original reference drugs (Remicade, Enbrel) and the almost identical copy drugs (biosimilars).

## 1. Introduction

Rheumatoid arthritis (RA) is thought to be generated by an autoimmune response in which the innate immune system (dendritic cells, macrophages) presents antigens to the adaptive immune system (T-cells, B-cells) initiating an inflammatory cascade, which enhances local inflammation of the synovial membrane (synovitis) and results in damage to the cartilage and bone components of the joints [1]. Over time, there is a strong association between the cumulated degree of synovitis and the development of joint damage [2,3,4]. Tumour necrosis factor (TNF) is a cytokine with an important role in this process, but the exact mechanism is still controversial [1]. The first TNF inhibitor (TNFi) shown to be effective in the treatment of RA is a chimeric monoclonal murine/human antibody to TNFα (cA2 = infliximab) directly inhibiting the TNF molecule [5], whereas the second is a genetically engineered TNF receptor blocker (etanercept) [6].

Three succeeding TNFis (adalimumab, certolizumab, and golimumab) are antibodies like infliximab, but are purely human. The original reference drugs are Remicade (infliximab), Enbrel (etanercept), Humira (adalimumab), Cimzia (certolizumab), and Simponi (golimumab). A detailed description of the structure of all five TNFis have recently been reviewed [1]. No studies have directly compared the efficacies of these inhibitors, but indirect evidence indicate that they are similar [7,8], although the effect of the defined standard dose of golimumab may be inferior [8]. This indirect evidence is based on randomized controlled trials (RCTs) comparing a TNFi in combination with the conventional disease modifying anti-rheumatic drug (DMARD), methotrexate (MTX), versus placebo plus MTX, or TNFi mono-therapy versus MTX mono-therapy. No RCT has compared a TNFi to an otherwise untreated placebo control.

In the recent years, statistical software has matured to combine such indirect comparisons into one integrated network analysis, based on which treatments can be ranked, and to include different doses of the target drugs in the analysis [9]. A PubMed search (20 April 2019) with the search term “meta-analysis and rheumatoid arthritis and TNF inhibitor” revealed 135 articles of which only 2 [10,11] are network meta-analyses, which exclusively compare the efficacies of the 5 known TNFis using the American College of Rheumatology improvement criteria (ACR 20, 50, and 70) and health assessment questionnaire (HAQ) scores as primary outcome measures. One network meta-analysis of 26 trials found that TNFi combined with MTX was superior to either MTX or TNFi alone. Increasing doses did not improve the efficacy [10]. Another analysis of 16 trials of MTX non-responders found that all TNFis showed considerably improved efficacy over placebo plus MTX but etanercept appeared superior to infliximab and golimumab, and certolizumab to infliximab and adalimumab [10]. One reason could be that the defined standard doses are not equivalent. Subsequently, a significant fraction of additional studies including studies of biosimilars, i.e., copy-products of the original reference drugs have appeared. As costs are one of the main obstacles for the use of TNFis, especially in low-income countries, a definition of the relative efficacy of the less expensive biosimilars is important.

The primary purpose of the present meta-analysis of randomized controlled trials (RCTs) was in one network to compare the standard doses of the five TNFis (reference drugs and biosimilars) combined with MTX to MTX mono-therapy (the primary comparator) and to TNFi mono-therapy, and for the first time also to an otherwise untreated placebo control, which was included in the network to give an estimation of the absolute inhibitory ability of the TNFis (assuming that placebo has no inhibitory effect). In contrast to the previous meta-analyses [10,11] we used radiographically estimated joint destruction [7,8,12] as the primary outcome. The secondary purpose was to relate TNFi doses below and above the standard doses to the standard doses and to MTX mono-therapy and placebo. The reference TNFis, as well as recent biosimilars, are included in the analysis. The analysis is reported in accordance with the Preferred Reporting Items for Systematic Reviews and Meta-Analyses incorporating network meta-analyses (PRISMA extension statement) [13].

## 2. Results

### 2.1. Result of the Search

The search resulted in 3184 articles. We organised the search in a Word document including all headlines and abstracts. To avoid overlooking studies with radiographic evaluation we searched the text for the term “radio” (which covers radiologic(al), radiology, radiograph(ic(al))) (1621 selections) and the term “X-ray” (131 selections). During this search, we excluded 2922 studies. We then read the 262 remaining abstracts in detail and excluded a further 111. Thus, 151 studies were included for retrieving full-length articles. From this pool of 151 articles we identified 36 RCTs (Figure 1), 31 TNFi RCTs and 5 RCTs, which could link an otherwise untreated placebo group to the network, 1 study of MTX vs. placebo, 3 studies of sulfasalazine vs. placebo, and 1 study of MTX vs. sulfasalazine. These 36 RCTs were published in 35 articles [14,15,16,17,18,19,20,21,22,23,24,25,26,27,28,29,30,31,32,33,34,35,36,37,38,39,40,41,42,43,44,45,46,47,48]. 

### 2.2. Description of Included Studies

Baseline characteristics are shown in Table 1. The number of patients in the studies varied from 24 to 1022 (Table 1, N column). The study duration varied from 24 weeks to 104 weeks (Table 1, SD column). The mean duration of disease (RA) varied from 0.2 to 11 years (Table 1, DD column). The mean age varied from 48.4 to 60.7 years (Table 1, Age column). The percentage of females varied from 64% to 85.7% (Table 1, F% column). The percentage of rheumatoid factor positive patients varied from 54.9% to 100% (Table 1, RF% column). The state of disease activity at baseline was reported by all studies. Most studies reported swollen joint count (SJC), tender joint count (TJC), and erythrocyte sedimentation rate (ESR) and/or C-reactive protein (CRP). These variables are included in the disease activity score based on 28 joints or 44 joints (DAS28 score or DAS44 score). Two studies reported DAS44 [17,25], which was converted to DAS28 as described in the methods section. In six studies with no report of DAS28 [14,15,16,18,19,23] we could estimate a mean DAS28 from SJC, TJC, and CRP or ESR as described in the methods section. Twenty-eight studies directly reported DAS28. Thus, a DAS28 value was available for all studies. DAS28 varied from 4.4 to 7.0 (Table 1, DAS28 column). The health assessment questionnaire score varied from 1.0 to 1.8 (Table 1, HAQ column). The state of chronicity is characterized by the radiographic score, which varied from 2 to 159 (Table 1, RS column).

Glucocorticoid (GC) was not allowed in two studies [17,20]. In the remaining studies the mean GC dose per patient varied from 0.4 mg to 4.8 mg (Table 1, GC column). One study directly reported the mean GC intake in the treatment groups [34]. Three studies did not report the GC treatment interval but did report the percentage of patients being treated with GC [21,40,43]. An estimated dose of 5 mg of GC multiplied by the percentage of treated patients was imputed for these three studies. Six studies only reported the interval of 0–10 mg of GC [32,36,39,45,47,48]. Assuming a median dose of 5 mg and 50% of the patients receiving GC, an estimated GC intake of 2.5 mg was imputed for these six studies. The remaining studies reported both the interval of intake and the percentage of patient taking steroids. The median GC dose of the interval multiplied by the percentage of treated patients was imputed for these studies.

### 2.3. Risk of Bias in Included Studies

We estimated bias by means of the Cochrane risk of bias tool [49]. Based on the description of the randomization procedure (Table 1, column RP) (sequence generation and concealed allocation), 11 studies featured sufficient description, and 25 studies insufficient description.

32 studies were double-blinded, whereas 4 studies were open studies [18,19,25,34], but the radiographic scores were estimated blindly in all studies. All but 3 studies [20,39,44] were initiated by pharmaceutical companies.

21 studies used the ‘intention to treat’ (ITT) principle with no use of data obtained during rescue therapy (i.e., open-label administration of alternative drug when the allocated treatment is insufficient) in the efficacy analyses, whereas 11 studies either used such data in the efficacy analyses or completely excluded drop-outs without using the ITT principle. In four studies, the use of ITT was unclear. Thus, there was no indication of incomplete outcome data (IOC) in 21 studies and indication of either obvious or unclear incomplete outcome data in 15 studies (Table 1, column IOC)

In 18 studies, the study populations had an inadequate response to previous MTX or other DMARD (DIR), whereas 18 study populations did not (Table 1, column DIR).

### 2.4. Trial Network

The trial network (Figure 2) is quite sparse with few direct comparisons available. Each node represents one treatment with shorthand labels, lines show direct comparisons in the included studies. Node sizes are proportional to number of patients receiving the treatments, line thicknesses to the number of studies with these comparisons. Treatments cluster according to active ingredient, and each cluster is well-connected to MTX. Biosimilars are compared only to their respective reference drugs, and Golimumab 130 mg combined with MTX (Go13Mt) to MTX only. Based on its high degree centrality [50] and its status as standard treatment, we chose MTX as the comparator to avoid uncertainty inflation due to network traversal. All fits showed satisfactory mixing and convergence of chains.

### 2.5. Treatment Effects

We performed 6 adjusted meta-regression fit analyses of the 25 treatments (including placebo) with each of the defined covariates individually-randomization procedure (dichotomous); incomplete outcome assessment (dichotomous); glucocorticoid treatment (continuous); disease activity score (continuous); health assessment score (continuous); total radiographic joint score, RS (continuous) (Table 2) and one including four covariates (Table 3). All continuous covariates were mean-centred and scaled to 1 standard deviation. Table 2 shows the unadjusted analysis and six regression analyses with one covariate at a time. The random-effects models, generally, yielded substantially lower DIC estimates than the corresponding fixed-effects models, suggesting superiority of the former. Thus, only the random-effects models are shown. The results are shown as difference in annual percent point between each of the investigated treatments, compared to MTX.

Adjustment for the randomization procedure covariate (Table 2, randomization column) decreased the treatment effects meaning that the effect of studies in which the randomization procedure was insufficiently described benefitted from this adjustment. A balance within the treatment groups between studies with sufficient and insufficient description of the randomization procedure would neutralize this effect, but such balance was not present, especially obvious for golimumab, as the randomization procedure was insufficiently described in all golimumab studies. Furthermore, this covariate shows the description of the randomization procedure, however, does not necessarily reflect how the randomization procedure was performed. We therefore excluded this variable from the multiple adjustment model.

Th RS model (Table 2, baseline RS column) had the lowest DIC (−50.4) and the largest coefficient (−0.3%) meaning that this model was the best to describe the data (lowest DIC value) and on the average changed the outcomes with −0.3%.

Due to the limited number of studies included it would not be meaningful with many covariates and we therefore also excluded the HAQ score (Table 2, HAQ column), as HAQ is a consequence of disease activity (Table 2, DAS28 column) and joint destruction (Table 2, baseline RS column), which accordingly were included together with GC treatment (Table 2, glucocorticoid dose column) and incomplete outcome (Table 2, incomplete outcome column).

Table 3 shows treatment effects, covariate coefficients, and diagnostic metrics for the random-effects of the unadjusted consistency model, the RS model and the four-covariate meta-regression model (normalized DAS28, normalized RS, normalized GC treatment and incomplete outcome). The fixed-effect model yielded similar results (not shown). The results are shown as difference in annual percent point between each of the investigated treatments compared to MTX.

The random-effects models, generally, yielded substantially lower DIC estimates and posterior mean residual deviances than the corresponding fixed-effects models, suggesting superiority of the former.

The adjusted models yielded a smaller estimate for σ (0.1; 95% CrI: 0.0, 0.2) than the unadjusted fit (Table 3, unadjusted model column), indicating that some of the heterogeneity can be explained by the covariates.

The RS model (Table 3, RS covariate metaregression column) has a lower DIC value (−50.4) than the four-covariate model (Table 3, 4-covariate metaregression column) (−47.6) indicating that the RS model is superior.

In general, the effects in the RS model are either improved or unchanged compared with the unadjusted model, whereas the four-covariate model does not add much to the effects except that the golimumab outcomes improve.

In the RS model and the four-covariate regression fit, the effect of the standard doses of infliximab (3 mg/kg/8 weeks ≈ 100 mg/4 weeks), adalimumab (40 mg/2 weeks = 80 mg/4 weeks), etanercept (50 mg/week = 200 mg/4 weeks) and certolizumab (200 mg/2 weeks = 400 mg/4 weeks) did not differ significantly from each other, varying from −0.6% to −1.0% per year compared with MTX (Table 3, values in bold type). These effects were −0.1% to −0.3% better than in the unadjusted model. The effects of the standard doses of the biosimilars (SB2, CTP, SB4) were identical to their reference drugs (Table 3). The RS adjustment did not improve the golimumab effects, but the adjustment by four covariates did render the effect of golimumab closer to the other TNFis (Table 3). Still, this effect remained statistically inferior (−0.4%). The effects of TNFi mono-therapy (vs. MTX) with standard doses varied from −0.1% to −0.5% (unadjusted), −0.2% to −0.6% (RS adjusted), and −0.2% to −0.9% (four-covariate adjusted), except certolizumab 200 mg (−1.3%). TNFi low-dose therapy combined with MTX had effects like the standard doses of TNFi monotherapy (−0.3% to −0.9%). Moderately high doses (infliximab 6 mg/kg/8 weeks and golimumab 130 mg/4 weeks) did not differ significantly from the standard doses, but extremely high infliximab doses (infliximab 10mg/kg/8 weeks and 20 mg/kg/8 weeks, 3.5 and 7 times the standard dose) seemed better than the standard doses (−1.1% and −1.4%). A cumulative rankogram (not shown) showed that the different doses of infliximab (reference drugs and biosimilars) generally constituted the top treatments judged from their surface under the cumulative ranking curves [51].

### 2.6. Additional Analyses

To explore the significance of MTX inadequate response we performed a stratified analysis comparing inadequate-responder to responder studies and found the following results for standard TNFi + MTX vs. MTX: Adalimumab: −0.7% vs. −0.4%; certolizumab: −0.7% vs. −0.3%; etanercept: −1% vs. −0.5%; golimumab: −0.2% vs. −0.1%; infliximab: −1.5% vs. −0.7%.

Some of the effects seemed not to fit into the pattern. The mono-treatment effect of certolizumab 200 mg is unusually large (−1.3%) and so are the differences between sulfasalazine, MTX, and placebo. Excluding the study of certolizumab 200 mg [41] did not change the remaining effect estimates. Excluding all sulfasalazine studies did not change the remaining results either.

Finally, we explored fitting a network meta-analysis model to the original RS scale outcome data with covariates for RS_max_ and study duration and found qualitatively similar results.

### 2.7. Inconsistency Checks

The network has nine potentially inconsistent loops (Table 4). Generally, the network and direct effect estimates agree well with each other, particularly for comparisons with greater support, i.e., with many patients in several studies. The node-splitting models did suggest inconsistency in the In6MT-In3Mt-Mt loop, seemingly stemming from consistent effect estimates of In3Mt being marginally (but not significantly) more effective than In6Mt (Table 4, row 7).

### 2.8. Model Fitness and Comparison

DIC values and posterior mean residual deviances, generally, favoured regression fits over unadjusted fits, and random-effects models over fixed-effects models.

## 3. Discussion

In many RA patients, mono-therapy and combination therapy with DMARDs are sufficient to neutralize inflammatory activity and inhibit radiographic progression [7,8]. Due to costs therapy with TNFis or other biologics should therefore only be given to RA patients with an insufficient response to combination therapy with DMARDs [7,8]. Still, to be able to estimate the absolute long-term effect it is important to define an estimate of the attainable effect of a TNFi, i.e., the total effect compared with no treatment (i.e., placebo treatment). The present network meta-analysis for the first time defines this effect to be about −0.9% for TNFi monotherapy and about −1.2% for a TNFi combined with MTX. However, the success of the attempt to include placebo in the network was limited, as the placebo studies included were poorly connected in the network and gave rise to borderline inconsistent results, which must be considered to be non-robust. Future studies with more power are needed to safely establish these effects.

The analysis also shows that that the effects of different TNFis on radiographic joint destruction in patients with RA generally are similar and that the effects of biosimilars correspond to the reference drugs. However, the effect of golimumab in the defined standard dose seemed to be inferior. Although there may be intraclass differences (high-dose TNFi vs. standard-dose TNFi vs. low-dose TNFi, and TNFi combination therapy vs. TNFi mono-therapy), the present analysis shows that these differences are clinically minor compared to the large difference between the TNFi effect and the placebo effect.

In the stratified analysis of MTX inadequate responders vs. MTX responders, the inadequate response group had −0.1 to −0.8 percent points better TNFi effects than the MTX responsive group, probably because the MTX effect is smaller in the inadequate response group. This further emphasizes that TNFi treatment primarily should be used in patients with insufficient response to MTX and other DMARDs. This analysis also indicates that, in future analyses, it may be appropriate to separate the MTX treatment in an adequate response group and an inadequate response group.

No statistically significant inconsistencies were detected (Table 4), but the effect of sulfasalazine, which was identical with the effect of MTX in the direct comparison, was −0.8% in the indirect comparison. However, in contrast to the direct comparison, 50% of the MTX studies included in the indirect comparison included MTX inadequate responders which could explain this inconsistency.

Thus, sulfasalazine was poorly connected to MTX (the central comparator) in the network. Accordingly, the high effect of monotherapy with certolizumab 200 mg, which had no direct connection to MTX in the network, could be explained by its link to the network through sulfasalazine. Paradoxically infliximab 3 mg/kg was borderline significantly better than infliximab 6 mg/kg in the indirect comparison, but infliximab 6 mg/kg was investigated in more MTX adequate responders (306) than inadequate responders (175), whereas Infliximab 3 mg/kg was investigated in fewer MTX adequate responders (480) than inadequate responders (534). This imbalance could explain the observed paradoxical difference.

In vitro, it has been demonstrated that golimumab has higher affinity, greater capacity to neutralize TNF and greater conformational stability than infliximab [52]. Such findings have entailed the assumption that lower doses of golimumab would be equivalent to the defined dose of infliximab. However, RCTs of golimumab have indicated that the effect of the standard dose of golimumab was inferior. Our network meta-analysis model without covariates gave results in line with these findings. Two golimumab studies were performed in MTX adequate responders and two in inadequate responders and in both groups the effect of golimumab were inferior compared with the other TNFis. After adjustment for covariates the effect of golimumab approached the effect of the other TNFis, but did not reach it, indicating that the definition of the standard dose of golimumab may be too low. This assumption is supported by the finding that the double standard golimumab dose (100 mg/4 weeks) reached the effect of the other TNFi standard doses.

Previous analyses used process variables, typically ACR response criteria [10,11], which are sensitive to current treatment. As the TNFis are administered with different time intervals, the outcome could be influenced by the time of measurement in the dosing cycle. Especially outcomes in patients treated with infliximab could be sensitive, as the dosing cycle of infliximab is 8 weeks and the evaluation of the patients typically are performed in the end of the cycle, which could explain the trend towards inferiority of infliximab [10,11]. The present analysis did not confirm this inferiority, maybe because we used the essential and approximately irreversible RA outcome, joint destruction, which is insensitive to short-term treatment variations.

### 3.1. Limitations

The premise that placebo has no effect may be wrong and, in that case, the true effects may be even bigger than the present reported effects versus placebo.

A review of RA controlled trials [53] showed that positive study outcome was associated with publication and timeliness of publication despite registration in a publicly available registry (Clinicaltrials.gov). A substantial number of RA-RCTs remained unpublished [53]. In their search for outcomes in unpublished studies, the authors had to rely on abstracts and terms as ‘unsuccessful’ and ‘negative’, which are enough to evaluate whether a study is positive or negative, but not sufficient for a meta-analysis, which demands detailed results. As the present meta-analysis mainly includes studies with positive outcomes, it is possible that publication bias may have an impact on the outcome and that the recorded effects may be higher than the true mean effects.

Linear extrapolation of radiographic scores were used in almost all studies to adjust radiographic data of participants, who dropped out before the final date of evaluation. We also used this method to standardize the response data, i.e., imposing a linearity assumption on disease progression. This assumption is supported by studies showing approximate linearity between disease duration and radiographic progression of joint destruction [54]. Furthermore, the sensitivity analysis comparing our standardized outcome with the original RS adjusted for duration and maximal RS showed similar results.

The GeMTC software package features automated identification of potentially inconsistent comparisons and subsequent analysis of these by way of node-splitting models for each of the identified comparisons [55] albeit with fixed vague priors beyond our control. The heterogeneity in the consistency model and the results of the node-splitting models did not suggest severe inconsistency, in which case there will not be global inconsistency either.

Heterogeneity and inconsistency are closely related, and we would not expect an inconsistency model to disclose problems that would undermine our findings.

### 3.2. Strengths

We have used WinBUGS models [9] widely recognized as valid in the community to elicit physiologically meaningful effect sizes of TNF inhibitors, with and without methotrexate.

In conclusion, this analysis shows that the effect of the defined standard doses of TNFis are close to the achievable efficacy and that only doses 3.5–7 times the standard dose contributed with a significant additional effect, albeit relatively small from a clinical point of view. Under normal conditions, such a dose would be unacceptable due to risk of side effects and for economic reasons. In contrast, below-standard doses of TNFis had effects significantly superior to MTX monotherapy, and therefore could be sufficient in many individuals with RA. Similarly, the fact that the effect of biosimilars corresponded to the effect of the reference TNFis suggests a potential for cutting costs of TNFi use.

## 4. Materials and Methods

### 4.1. Types of Studies

Full-length studies published in peer-reviewed journals that were performed according to a randomized controlled trial design and that scored joint radiographs as the primary or secondary outcome at two separate time points with a time interval of at least 3 months were included, irrespective of sample size and publication year.

### 4.2. Types of Participants

Patients with RA diagnosed according to the 1958 [56] or the 1987 [57] or the 2010 [58] criteria of the American College of Rheumatology (ACR; formerly, the American Rheumatism Association) were included. In studies performed before 1959, the stated study definitions of RA were accepted.

### 4.3. Types of Interventions

All interventions, which included a TNFi either as mono-therapy or in combination with another drug and compared with a DMARD or different dose of the same drug or biosimilar were selected for the present analysis. Furthermore, we selected studies with an otherwise untreated placebo-arm, which through its comparator could be linked to the network of TNFi treatments to be able to indirectly compare the efficacy of TNFis with an otherwise untreated placebo group.

### 4.4. Type of Outcome

The outcome was primarily recorded as the change in radiographic score (RS) measured as total Sharp score or a modification of total Sharp score [12] or Larsen score [54] between baseline and time of evaluation. To obtain our outcome variable we then standardized these to the percentage of the annual radiographic progression rate (PARPR) [4] by multiplying the mean score change and its standard error with a factor K = (100% × 52 weeks)/(RS_max_ × study duration weeks). This transformation is justified as radiographic progression and has been shown to be approximately linear [54]

### 4.5. Search Method for Identification of Studies

All mono-therapy or combination-therapy interventions including small molecule or biologic disease-modifying antirheumatic drugs were included in the search. These drugs interventions were 1) DMARDs (leflunomide, MTX, sulfasalazine, injectable gold, chloroquine, cyclosporin A, D-penicillamine, oral gold, azathioprine, cyclophosphamide, and tacrolimus); 2) glucocorticoids; 3) biologic agents (tumour necrosis factor antagonists (infliximab, etanercept, adalimumab, certolizumab, golimumab), anti-CD20 antibody (rituximab, ocrelizumab, ofatumumab), T cell co-stimulation inhibitor (abatacept), interleukin-6 (IL-6) antagonists (tocilizumab, sarilimumab), IL-1 receptor antagonist (anakinra); and 4) Janus Kinase Inhibitors (tofacitinib, baricitinib). For the present study focusing on the effect of TNFis, TNFi studies are selected from this pool together with studies, which could link the TNFi effect to an otherwise untreated placebo effect.

The search strategy was designed by two authors (LG and NG). We performed the last search 20 April 2019, using the following search strategy in PubMed:

“rheumatoid arthritis and controlled trial and chloroquine OR rheumatoid arthritis and controlled trial and azathioprine OR rheumatoid arthritis and controlled trial and penicillamine OR rheumatoid arthritis and controlled trial and cyclophosphamide OR rheumatoid arthritis and controlled trial and methotrexate OR rheumatoid arthritis and controlled trial and sulfasalazine OR rheumatoid arthritis and controlled trial and leflunomide OR rheumatoid arthritis and controlled trial and gold OR rheumatoid arthritis and controlled trial and cyclosporine OR rheumatoid arthritis and controlled trial and infliximab OR rheumatoid arthritis and controlled trial and etanercept OR rheumatoid arthritis and controlled trial and adalimumab OR rheumatoid arthritis and controlled trial and certolizumab OR rheumatoid arthritis and controlled trial and golimumab OR rheumatoid arthritis and controlled trial and tocilizumab OR rheumatoid arthritis and controlled trial and abatacept OR rheumatoid arthritis and controlled trial and rituximab OR rheumatoid arthritis and controlled trial and ocrelizumab OR rheumatoid arthritis and controlled trial and ofatumumab OR rheumatoid arthritis and controlled trial and glucocorticoid OR rheumatoid arthritis and controlled trial and cortisone OR rheumatoid arthritis and controlled trial and tofacitinib OR rheumatoid arthritis and controlled trial and baricitinib OR rheumatoid arthritis and controlled trial and anakinra OR rheumatoid arthritis and controlled trial and sarilimumab OR rheumatoid arthritis and controlled trial and tacrolimus“.

### 4.6. Study Selection

Two authors (NG and THG) independently assessed studies for eligibility. Disagreements were resolved by consensus. Titles were screened, possible abstracts were read, and possible papers were retrieved. Studies fulfilling the eligibility criteria were included in the meta-analysis. There were no exclusion criteria for studies fulfilling all inclusion criteria.

### 4.7. Data Extraction and Management

Two authors (NG/GJ and NG/THG) independently extracted data as published in the original articles. Disagreements were resolved by discussion between the authors. As a rule, the first publication of studies with repeated publications of follow-up data was included, but additional publications were used to provide missing data if possible. Data were recorded on a standardized extraction form using Excel spreadsheet software from Microsoft. The following baseline variables for each treatment arm were recorded: mean age, female percentage, percent rheumatoid factor positivity, tender joint count (TJC), swollen joint count (SJC), erythrocyte sedimentation rate (ESR), C-reactive protein (CRP), disease activity score based on 28 joints (DAS28) or 44 joint (DAS44), health assessment questionnaire score (HAQ) and radiographic score (RS: the sum of erosion score and joint narrowing score). Not all studies provided data with identical units. We therefore converted some measures from some studies to the most prevailing unit.

DAS28 conversion: Most of the studies informed about DAS28, but many did not report whether it was based on 3 or 4 items or on CRP or ESR. However, according to a systematic comparison of composite measures [59] these 4 DAS28 values are similar at the 50th centile (i.e., the median) and we therefore recorded the DAS28 values as published in the original articles. A few studies published DAS44 values (based on 44 joints). These were changed to estimated DAS28 values by means of the nomogram in the paper by Ranganath et al. [58]. In studies with no report of DAS28 we could estimate a mean DAS28 from SJC, TJC, and ESR.

Conversion of SJC28 to SJC66: Three of the included studies measured both, and the SJC28 count in the six treatment arms varied between 67% and 74% of the SJC66 count. These numbers correspond well to the numbers estimated by Fuchs and Pincus [60]. They investigated systematically the individual participant data from three randomized trials and found these percentages to be 68, 69, and 83 (mean value 73%). We chose to use this value (73%) as a conversion factor between SJC28 and SJC66.

Conversion of TJC28 to TJC68: Three of the included studies measured both, and the TJC28 count in the six treatment arms varied between 55% and 64% of the TJC68 count. These numbers correspond to the numbers estimated by Fuchs and Pincus [60], who found these percentages to be 57, 60, and 71 (mean value 63%). We chose to use this value (63%) as a conversion factor between TJC28 and TJC68.

### 4.8. Assessment of Risk of Bias in Included Studies

The quality of the studies that were included in our meta-analysis was ensured by requiring randomization and by using the Cochrane risk of bias tool to assess sequence generation, allocation concealment, blinding, and incomplete outcome data at the level of the individual study [49]. In addition, blinding was evaluated at the outcome level. The analysis of incomplete outcome data included recording of the following features: 1) dropout frequency (total number of patients dropping out after randomization and first treatment due to lack of effect or side effects, 2) change of treatment strategy (change of DMARD, addition of DMARD or glucocorticoids, change of placebo to DMARD), and 3) use of intention-to-treat principle. In addition, we recorded the use glucocorticoids.

### 4.9. Measure of Treatment Effect

This was defined as the mean difference (MD) between the change in RS from baseline to time of evaluation. The uncertainty of the observed treatment effect was quantified as the MD standard error.

### 4.10. Missing Data

We evaluated whether there was bias due to missing studies (publication bias), missing outcomes (selective reporting bias), or missing individuals (loss of participants during study: attrition bias).

If the SE was not reported it was calculated from a given SD, 95% confidence interval (CI), P value or *t*-value, estimated from a figure or imputed from the formula SE (change) = sq. root (SE1sq + SE2sq), SE1 is SE on R score before intervention and SE2 is SE on R score after intervention.

### 4.11. Data Synthesis

Outcome and covariates for meta-regression: the extracted outcome estimates varied in three aspects: the radiographic scoring technique used, and joints included, and follow-up time. To rein these differences, we derived a conversion factor K = (100% × 52 [weeks])/(RS_max_ × study duration [weeks]). Multiplying the RS mean differences and standard errors by K, we obtained annual progressions in percent and corresponding standard errors [61]. When estimating the absolute treatment effects, we calculated the weighted mean effect of MTX to follow an N (0.75, 0.08^2^) distribution, arising from a weighted average of the observed absolute treatment effect of MTX in the included studies.

We extracted data for six study-level covariates for the meta-regressions: average glucocorticoid dose(mg) used by patients, standardized to prednisolone dose; baseline average HAQ score (continuous); baseline average DAS28 score (continuous); baseline average total RS (continuous); description of randomization procedure (sufficient vs. insufficient) assessed on the basis of the description of sequence generation and allocation concealment; indicator of whether the outcome was incomplete (no vs. yes or unclear). An outcome was defined as complete if the data-analysis was performed according to the intention to treat principle under the condition that only data obtained during the study treatment was used in the final analysis by using the last observation carried forward or linear extrapolation of the last observation in case of study drug discontinuation. An outcome was defined as incomplete if only participants completing the study were evaluated, or if intention to treat data obtained from participants after discontinuation of study treatment and change to another treatment were included in the data-analysis. All continuous covariates were mean-centred and scaled to 1 standard deviation.

### 4.12. Model Specification and Parameter Estimation

We used only open source software: Analyses and visualization were done with the R statistical programming language [R Core Team (2018). R: A language and environment for statistical computing. R Foundation for Statistical Computing, Vienna, Austria. URL available online: https://www.R-project.org/] version 3.5.1. invoking WinBUGS with the R2WinBUGS Package [62] 

We built and ran several fixed-effects and random-effects models with the heterogeneity parameter σ shared by all treatments: unadjusted consistency model of the treatment effect as difference in annual progression rate between the treatment in question and MTX [63]; six single-covariate consistency meta-regression analyses to identity which of the six covariates stated above had substantial impact on the treatment effects; based on this ‘screening’, we selected the most appropriate covariates to be included in a multiple regression analysis. The models were linear with Gaussian likelihoods. We assumed covariate effects to be shared by all treatments, likely rendering the results more useful for subsequent decision making [64].

To explore the significance of MTX inadequate response, we performed a stratified analysis comparing inadequate MTX responders vs. MTX responders, using the random-effects multiple regression model.

We used the R package GeMTC [65] to identify potentially inconsistent loops, and automatically run the appropriate node-splitting models to gauge the differences between direct, indirect, and collective network estimates of treatment effects using vague priors akin to those we used in the WinBUGS models [58].

Unadjusted models had 50,000 sampling iterations and regressions models 100,000 for each of the four chains, keeping every 20th posterior sample to minimize auto-correlation and ensure precise estimation. All models ran 50,000 warm-up iterations. To ascertain convergence, we required potential scale reduction factors ≤ 1.05 [66], overlain posterior densities and mixing chain traces. We gauged model fitness with the deviance information criterion (DIC) and posterior mean residual deviance: lower DIC values are preferred, with differences between DICs of 3—5 being meaningful; the posterior mean residual deviance should be seen in light of the number of data points and should not go up when more parameters are added. We used vague N(0, 100^2^) priors for nuisance parameters, treatment effects, and covariate coefficients [63].

## Figures and Tables

**Figure 1 ijms-20-04350-f001:**
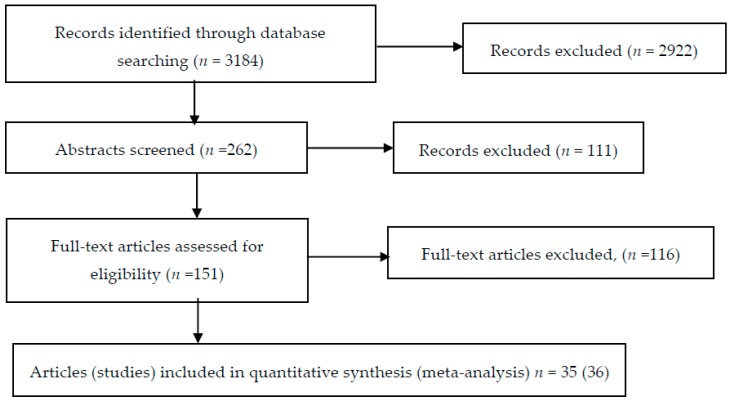
Flow diagram of search.

**Figure 2 ijms-20-04350-f002:**
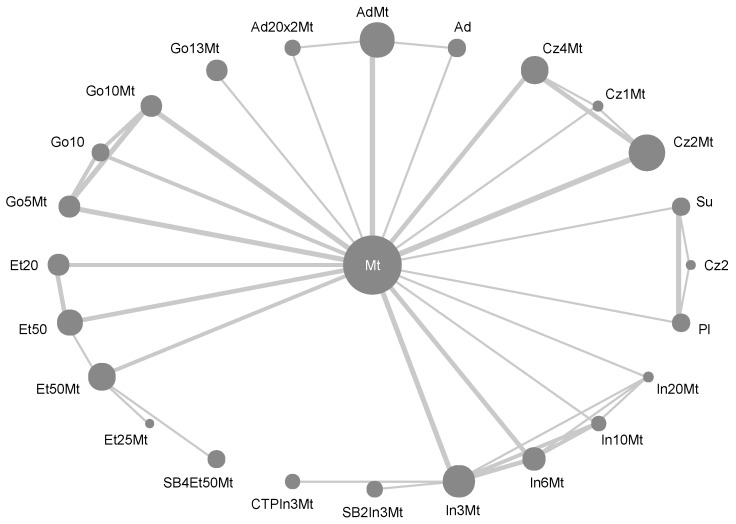
Network of treatments investigated in included studies. (Ad20x2Mt (adalimumab 20 mg/1 week plus MTX); AdMt (adalimumab 40 mg/2 weeks plus MTX); Ad (adalimumab 40 mg/2 weeks); Cz4Mt (certolizumab 400 mg/2 weeks plus MTX); Cz1Mt (certolizumab 100 mg/2 weeks plus MTX); Cz2Mt (certolizumab 200 mg/2 weeks plus MTX); Cz2 (certolizumab 200 mg/2 weeks); Su (sulfasalazine); Pl (placebo); In20Mt (infliximab 20 mg/kg/8 weeks plus MTX); In10Mt (infliximab 10 mg/kg/8 weeks plus MTX); In6Mt (infliximab 6 mg/kg/8 weeks plus MTX); In3Mt (infliximab 3 mg/kg/8 weeks plus MTX); SB2 (biosimilar infliximab); CTP (biosimilar infliximab); SB4 (biosimilar etanercept); Et25Mt (etanercept 25 mg/1 week plus MTX); Et50Mt (etanercept 50 mg/1 week plus MTX); Et50 (etanercept 50 mg/1 week); Et20 (etanercept 20 mg/1 week); Go5Mt (golimumab 50 mg/4 weeks plus MTX); Go10 (golimumab 100 mg/4 weeks); Go10Mt (golimumab 100 mg/4 weeks plus MTX); Go13Mt (golimumab 130 mg/4 weeks plus MTX)).

**Table 1 ijms-20-04350-t001:** Baseline study characteristics.

Ref.	TD/C	GC. mg	DIR	RP	IOC	N	SD, weeks	DD, years	Age, years	F%	RF%	DAS28	HAQ	RS
[14]	Su/Pl	0.4	No	IS	Yes	73	52	0.5	51.3	64.0	66.7	5.6	1.4	2.0
[15]	Su/Pl	1.3	No	S	No	137	24	6.6	58.9	72.5	81.5	6.8	1.1	57.6
[16]	Mt/Pl	2.7	No	S	No	221	52	6.7	54.0	72.8	59.8	6.4	1.3	24.1
[17]	Mt/Su	0.0	No	IS	No	95	52	1.2	51.0	72.5	68.5	5.5	1.3	7.2
[18]	In/Mt	3.2	Yes	IS	Yes	135	54	10.5	52.5	80.5	80.5	6.7	1.8	80.5
[19]	Et/Mt	2.8	No	IS	Yes	390	52	1.0	49.5	75.0	88.5	6.9	1.4	12.1
[20]	Su/Pl	0.0	No	IS	Yes	83	52	0.5	57.1	75.0	56.0	5.2	1.2	3.9
[21]	Et/Mt	3.1	Yes	IS	UC	424	52	6.6	53.1	78.0	73.0	5.6	1.4	32.2
[22]	In/Mt	1.9	No	S	No	532	54	0.9	50.5	73.0	71.0	6.7	1.5	11.5
[23]	Ad/Mt	2.6	Yes	IS	Yes	368	52	11.0	56.7	74.3	85.4	6.1	1.5	66.5
[24]	In/Mt	2.5	Yes	S	No	24	54	1.5	53.3	75.5	100.0	5.3	1.4	18.0
[25]	In/Mt	0.0	No	S	Yes	234	52	0.6	54.0	67.0	65.5	5.5	1.4	7.2
[26]	Ad/Mt	2.2	No	IS	Yes	531	52	0.8	52.1	75.7	79.9	6.4	1.6	20.4
[27]	Et/Mt	2.5	No	S	No	476	52	0.8	51.4	73.5	68.5	6.5	1.7	6.0
[28]	Cz/Mt	1.7	Yes	IS	No	592	52	6.2	51.8	83.2	81.2	7.0	1.7	41.1
[29]	Cz/Mt	2.8	Yes	IS	No	373	24	5.9	51.9	84.0	77.9	6.8	1.6	43.1
[30]	In/In	3.4	Yes	IS	No	178	40	7.8	49.3	80.8	87.0	6.2	1.2	48.5
[31]	Go/Mt	3.3	No	IS	No	280	52	3.5	48.4	84.1	81.5	5.1	1.6	20.1
[31]	Go/Mt	4.8	Yes	IS	No	234	52	6.2	51.5	80.5	82.4	6.1	1.4	37.1
[32]	Go/Mt	2.5	Yes	IS	Yes	165	24	8.8	50.8	84.0	79.9	5.6	1.0	56.1
[33]	Ad/Mt	2.2	No	IS	No	1022	26	0.4	50.6	74.0	88.0	6.0	1.6	11.5
[34]	Et/Et	4.4	Yes	IS	UC	62	52	9.2	60.7	80.0	81.4	4.8	1.0	159.9
[35]	Et/Et	3.8	No	IS	No	63	104	10.9	60.6	85.7	75.0	4.4	1.4	54.0
[36]	Go/Mt	2.5	Yes	IS	No	592	24	7.0	51.7	81.1	100.0	6.0	1.6	49.0
[37]	Et/Mt	3.0	Yes	S	No	361	52	3.0	51.0	79.9	75.6	5.8	1.1	44.1
[38]	Go/Mt	2.5	Yes	IS	Yes	206	24	8.7	52.7	81.1	NR	5.9	1.1	50.0
[39]	In/Mt	1.6	No	S	No	112	26	0.1	53.3	68.7	54.9	4.3	1.4	7.6
[40]	Ad/Mt	1.6	No	IS	No	331	26	0.3	54.0	81.3	84.4	6.6	1.2	13.7
[41]	Cz/Su.Pl	3.6	Yes	S	No	114	24	5.8	55.4	77.2	89.5	6.3	1.2	46.1
[42]	Cz/Mt	3.1	Yes	S	No	146	24	5.9	53.1	83.2	88.0	6.4	1.2	53.8
[43]	Cz/Mt	0.9	No	IS	UC	315	24	0.3	49.2	81.0	96.2	5.5	1.1	5.6
[44]	Ad/Mt	0.9	No	IS	Yes	173	52	0.2	55.2	66.0	72.0	5.6	1.1	4.4
[45]	In/In	2.5	Yes	IS	UC	336	54	NR	50.0	82.7	73.5	5.9	1.6	66.6
[46]	Cz/Mt	1.6	No	IS	No	691	52	0.3	50.8	77.9	96.8	6.8	1.7	7.9
[47]	Et/Et	2.5	Yes	IS	Yes	478	52	6.1	51.9	84.3	78.6	6.5	1.5	41.1
[48]	In/In	2.5	Yes	S	No	422	54	6.5	52.1	80.1	72.5	6.5	1.5	38.0

TD/C: testdrug/control; Ad: adalimumab; Cz: certolizumab; Et: etanercept; Go: golimumab; In: infliximab; Mt: methotrexate; Pl: placebo; Su: sulfasalazine; GC: glucocorticoid (prednisolone equivalent); DIR: DMARD inadequate response; RP: randomization procedure; IS: insufficiently described; S: sufficiently described; IOC: incomplete outcome; UC: unclear; SD: study duration; DD: disease duration; F: female; RF: rheumatoid factor; DAS28: disease activity score based on evaluation of 28 joints; HAQ: health assessment score; RS: total radiographic joint score.

**Table 2 ijms-20-04350-t002:** Random-effects regression analyses with one covariate at a time.

Treatment #	Unadjusted	DAS28 *	HAQ *	Baseline RS *	Glucocorti-Coid Dose *	Incomplete Outcome	Randomi-Zation Procedure
Coefficient	—	−0.0 (−0.2, 0.1)	0.0 (−0.1, 0.2)	−0.3 (−0.4, 0.1)	−0.1 (−0.2, 0.1)	−0.1 (−0.3, 0.2)	0.2 (−0.1, 0.6)
Ad40	−0.1 (−0.8, 0.5)	−0.1 (−0.8, 0.5)	−0.1 (−0.8, 0.5)	−0.2 (−0.8, 0.3)	−0.2 (−0.8, 0.4)	−0.1 (−0.8, 0.5)	−0.4 (−1.1, 0.4)
Ad20x2Mt	−0.4 (−0.8, 0.0)	−0.4 (−0.8, 0.0)	−0.4 (−0.8, 0.0)	−0.3 (−0.6, 0.0)	−0.4 (−0.8, 0.0)	−0.3 (−0.8, 0.1)	−0.6 (−1.2, −0.1)
Ad40Mt	−0.5 (−0.7, −0.3)	−0.5 (−0.7, −0.3)	−0.5 (−0.7, 0.3)	−0.6 (−0.8, 0.4)	−0.5 (−0.8, 0.3)	−0.5 (−0.7, 0.2)	−0.7 (−1.2, −0.3)
Cz100Mt	−0.2 (−0.9, 0.3)	−0.2 (−0.8, 0.4)	−0.3 (−0.9, 0.3)	−0.3 (−0.8, 0.2)	−0.3 (−0.9, 0.3)	−0.2 (−0.9, 0.4)	−0.4 (−1.2, 0.2)
Cz200	−1.1 (−2.1, −0.2)	−1.1 (−2.1, −0.1)	−1.1 (−2.1, 0.1)	−1.3 (−2.1, 0.4)	−1.1 (−2.1, 0.1)	−1.1 (−2.1, 0.1)	−1.1 (−2.1, −0.2)
Cz200Mt	−0.5 (−0.7, −0.3)	−0.4 (−0.7, −0.2)	−0.5 (−0.8, 0.3)	−0.6 (−0.8, 0.4)	−0.5 (−0.8, 0.3)	−0.5 (−0.7, 0.2)	−0.7 (−1.1, −0.3)
Cz400Mt	−0.5 (−0.9, −0.3)	−0.5 (−0.8, −0.2)	−0.6 (−0.9, 0.3)	−0.6 (−0.9, 0.4)	−0.6 (−0.9, 0.3)	−0.5 (−0.9, 0.3)	−0.8 (−1.2, −0.3)
Et20	−0.2 (−0.6, 0.1)	−0.2 (−0.6, 0.1)	−0.2 (−0.6, 0.1)	−0.3 (−0.6, 0.0)	−0.2 (−0.6, 0.1)	−0.2 (−0.6, 0.2)	−0.4 (−0.8, 0.0)
Et25Mt	−0.5 (−1.1, 0.0)	−0.5 (−1.1, 0.0)	−0.6 (−1.2, 0.0)	−0.7 (−1.2, 0.3)	−0.5 (−1.1, 0.0)	−0.5 (−1.2, 0.1)	−0.6 (−1.3, −0.0)
Et50	−0.5 (−0.8, −0.2)	−0.5 (−0.8, −0.2)	−0.5 (−0.8, 0.2)	−0.6 (−0.9, 0.3)	−0.4 (−0.8, 0.2)	−0.4 (−0.8, 0.1)	−0.6 (−1.0, −0.2)
SB4Et50Mt	−0.7 (−1.2, −0.2)	−0.7 (−1.2, −0.2)	−0.7 (−1.3, −0.2)	−0.9 (−1.2, −0.5)	−0.7 (−1.2, 0.2)	−0.7 (−1.2, −0.1)	−0.8 (−1.3, −0.3)
Et50Mt	−0.6 (−0.9, −0.3)	−0.6 (−0.9, −0.3)	−0.7 (−1.0, −0.3)	−0.8 (−1.1, −0.5)	−0.6 (−0.9, 0.3)	−0.6 (−0.9, −0.3)	−0.7 (−1.1, −0.4)
Go100	−0.1 (−0.4, 0.2)	−0.1 (−0.5, 0.2)	−0.1 (−0.5, 0.2)	−0.2 (−0.4, 0.1)	−0.0 (−0.4, 0.3)	−0.1 (−0.4, 0.2)	−0.4 (−0.9, 0.1)
Go50Mt	−0.2 (−0.5, 0.1)	−0.2 (−0.5, 0.1)	−0.2 (−0.5, 0.1)	−0.2 (−0.4, 0.0)	−0.1 (−0.5, 0.2)	−0.2 (−0.5, 0.1)	−0.4 (−0.9, 0.0)
Go100Mt	−0.4 (−0.6, −0.1)	−0.4 (−0.7, −0.1)	−0.4 (−0.6, −0.1)	−0.4 (−0.6, −0.2)	−0.3 (−0.6, 0.0)	−0.3 (−0.6, −0.1)	−0.6 (−1.1, −0.1)
Go130Mt	−0.5 (−0.9, −0.1)	−0.5 (−1.0, −0.1)	−0.6 (−1.1, −0.1)	−0.4 (−0.8, −0.1)	−0.5 (−1.0, 0.1)	−0.5 (−1.0, −0.1)	−0.7 (−1.3, −0.2)
SB2In3Mt	−0.9 (−1.4, −0.4)	−0.9 (−1.4, −0.4)	−0.9 (−1.5, −0.4)	−1.0 (−1.4, −0.7)	−0.9 (−1.4, 0.4)	−0.9 (−1.4, −0.4)	−0.9 (−1.4, −0.4)
CTPIn3Mt	−0.8 (−1.3, −0.3)	−0.8 (−1.4, −0.3)	−0.8 (−1.4, −0.3)	−0.9 (−1.4, −0.5)	−0.8 (−1.4, 0.3)	−0.8 (−1.4, −0.2)	−0.8 (−1.4, −0.3)
In3Mt	−0.9 (−1.2, −0.6)	−0.9 (−1.2, −0.6)	−0.9 (−1.2, −0.6)	−1.0 (−1.3, −0.8)	−0.9 (−1.2, 0.6)	−0.9 (−1.2, −0.6)	−0.9 (−1.2, −0.6)
In6Mt	−0.8 (−1.2, −0.5)	−0.8 (−1.2, −0.5)	−0.9 (−1.2, −0.5)	−0.9 (−1.2, −0.7)	−0.9 (−1.2, 0.5)	−0.8 (−1.2, −0.5)	−0.9 (−1.2, −0.5)
In10Mt	−1.0 (−1.4, −0.7)	−1.0 (−1.4, −0.7)	−1.1 (−1.5, −0.7)	−1.1 (−1.4, −0.8)	−1.1 (−1.5, 0.7)	−1.0 (−1.4, −0.6)	−1.1 (−1.5, −0.7)
In20Mt	−1.3 (−1.8, −0.9)	−1.3 (−1.8, −0.9)	−1.4 (−1.9, −0.9)	−1.4 (−1.8, −1.0)	−1.3 (−1.8, 0.9)	−1.3 (−1.8, −0.8)	−1.4 (−1.9, −0.9)
Sulfasalazin	−0.7 (−1.2, −0.1)	−0.7 (−1.2, −0.1)	−0.7 (−1.2, −0.0)	−0.8 (−1.3, −0.3)	−0.7 (−1.3, 0.1)	−0.7 (−1.2, −0.1)	−0.7 (−1.3, −0.1)
Placebo	0.4 (−0.0, 0.8)	0.4 (−0.0, 0.8)	0.4 (−0.0, 0.8)	0.2 (−0.1, 0.6)	0.4 (−0.0, 0.8)	0.4 (−0.1, 0.8)	0.4 (−0.1, 0.8)
σ **	0.2 (0.1, 0.3)	0.2 (0.1, 0.3)	0.2 (0.1, 0.3)	0.1 (0.0, 0.2)	0.2 (0.1, 0.3)	0.2 (0.1, 0.3)	0.2 (0.1, 0.3)
Dbar †	−117.4	−117.4	−119.7	−119.5	−117.4	−118.2	−119.1
DIC	−43.0	−42.1	−44.0	−50.4	−42.3	−42.6	−43.4
pD	74.3	75.3	75.7	69.0	75.0	75.5	75.8

#: all treatment compared to MTX; * Scaled to 0 mean and 1 standard deviation; **: heterogeneity; † Compared to 93 data points from 36 studies; Ad: adalimumab; Cz: certolizumab; Et: etanercept; Go: golimumab; In: infliximab; Mt: methotrexate; SB2 (infliximab biosimilar), CTP (infliximab biosimilar) and SB4 (etanercept biosimilar) are biosimilar products. Dbar: posterior mean residual deviance; DIC: deviance information criterion; pD: estimated effective number of parameters.

**Table 3 ijms-20-04350-t003:** Difference in annual percent point progression between each of the investigated treatments, compared to methotrexate. Positive values favour methotrexate. Values are median of posterior distribution (95% CrI).

Treatment #	Four-Covariate Metaregression	RS Covariate Metaregression	Unadjusted Model
	Random Eff.	Random Eff.	Random Eff.
DAS28*	0.0 (−0.1, 0.1)	-	-
Incomplete outcome	0.1 (−0.1, 0.3)	-	-
Glucocorticoid dose*	0.1 (−0.1, 0.3)	-	-
Relative base RS*	−0.3 (−0.5, −0.2)	−0.3 (−0.4, 0.1)	-
Ad40	−0.2 (−0.8, 0.3)	−0.2 (−0.8, 0.3)	−0.1 (−0.8, 0.5)
Ad20x2Mt	−0.3 (−0.7, 0.0)	−0.3 (−0.6, 0.0)	−0.4 (−0.8, 0.0)
Ad40Mt	**−0.6 (−0.9, −0.4)**	**−0.6 (−0.8, 0.4)**	**−0.5 (−0.7, −0.3)**
Cz100Mt	−0.3 (−0.9, 0.2)	−0.3 (−0.8, 0.2)	−0.2 (−0.9, 0.3)
Cz200	−1.3 (−2.2, −0.4)	−1.3 (−2.1, 0.4)	−1.1 (−2.1, −0.1)
Cz200Mt	**−0.6 (−0.8, −0.4)**	**−0.6 (−0.8, 0.4)**	**−0.5 (−0.7, −0.3)**
Cz400Mt	−0.6 (−0.9, −0.4)	−0.6 (−0.9, 0.4)	−0.5 (−0.8, −0.3)
Et20	−0.5 (−0.9, −0.1)	−0.3 (−0.6, 0.0)	−0.2 (−0.6, 0.1)
Et25Mt	−0.9 (−1.4, −0.3)	−0.7 (−1.2, 0.3)	−0.5 (−1.1, 0.0)
Et50	−0.8 (−1.1, −0.4)	−0.6 (−0.9, 0.3)	−0.5 (−0.8, −0.2)
SB4Et50Mt	**−1.0 (−1.4, −0.6)**	**−0.9 (−1.2, −0.5)**	**−0.7 (−1.2, −0.2)**
Et50Mt	**−0.9 (−1.3, −0.6)**	**−0.8 (−1.1, −0.5)**	**−0.6 (−0.9, −0.3)**
Go100	−0.4 (−0.8, 0.0)	−0.2 (−0.4, 0.1)	−0.1 (−0.4, 0.2)
Go50Mt	**−0.4 (−0.8, −0.0)**	**−0.2 (−0.4, 0.0)**	**−0.2 (−0.5, 0.1)**
Go100Mt	−0.6 (−0.9, −0.2)	−0.4 (−0.6, −0.2)	−0.4 (−0.6, −0.1)
Go130Mt	−0.4 (−0.7, −0.1)	−0.4 (−0.8, −0.1)	−0.5 (−1.0, −0.1)
SB2In3Mt	**−1.0 (−1.4, −0.6)**	**−1.0 (−1.4, −0.7)**	**−0.9 (−1.4, −0.4)**
CTPIn3Mt	**−0.9 (−1.4, −0.5)**	**−0.9 (−1.4, −0.5)**	**−0.8 (−1.3, −0.3)**
In3Mt	**−1.0 (−1.3, −0.7)**	**−1.0 (−1.3, −0.8)**	**−0.9 (−1.2, −0.6)**
In6Mt	−0.9 (−1.3, −0.7)	−0.9 (−1.2, −0.7)	−0.8 (−1.2, −0.5)
In10Mt	−1.1 (−1.5, −0.8)	−1.1 (−1.4, −0.8)	−1.0 (−1.4, −0.7)
In20Mt	−1.4 (−1.8, −1.0)	−1.4 (−1.8, −1.0)	−1.3 (−1.8, −0.9)
Sulfasalazin	−0.8 (−1.4, −0.3)	−0.8 (−1.3, −0.3)	−0.7 (−1.2, −0.1)
Placebo	0.2 (−0.2, 0.5)	0.2 (−0.1, 0.6)	0.4 (−0.0, 0.8)
σ **	0.1 (0.0, 0.2)	0.1 (0.0, 0.2)	0.2 (0.1, 0.3)
Dbar †	−119.4	−119.5	−117.4
DIC	−47.6	−50.4	−43.0
pD	71.8	69.0	74.3

Effect of standard TNFi doses are in bold. # All treatment compared to MTX; * Scaled to 0 mean and 1 standard deviation; ** Heterogeneity; † Compared to 93 data points from 36 studies. Ad: adalimumab; Cz: certolizumab; Et: etanercept; Go: golimumab; In: infliximab; Mt: methotrexate; SB2 (infliximab), CTP (infliximab), and SB4 (etanercept) are biosimilar products. Dbar: posterior mean residual deviance; DIC: deviance information criterion; pD: estimated effective number of parameters.

**Table 4 ijms-20-04350-t004:** Results of node-splitting analysis.

Comparison	Consistency Effect	Direct Effect	Indirect Effect	*p*-Value
Et50Mt vs. Et50	−0.2 (−0.5, 0.2)	−0.2 (−0.7, 0.2)	0.1 (−0.5, 0.7)	0.42
Mt vs. Et20	0.2 (−0.1, 0.6)	0.3 (−0.1, 0.7)	0.1 (−0.5, 0.8)	0.71
Mt vs. In10Mt	1.0 (0.7, 1.4)	1.5 (0.8, 2.2)	0.8 (0.4, 1.3)	0.12
Mt vs. In3Mt	0.9 (0.6, 1.2)	0.8 (0.5, 1.1)	2.1 (0.7, 3.5)	0.08
Mt vs. In6Mt	0.8 (0.5, 1.2)	0.9 (0.5, 1.3)	0.6 (−0.0, 1.3)	0.40
In6Mt vs. In3Mt	0.1 (−0.2, 0.3)	0.1 (−0.2, 0.3)	−1.2 (−2.6, 0.2)	0.08
Pl vs. Mt	0.4 (−0.0, 0.8)	0.3 (−0.1, 0.7)	1.1 (−0.3, 2.6)	0.28
Su vs. Mt	−0.7 (−1.2, −0.1)	0.0 (−1.3, 1.4)	−0.8 (−1.4, −0.2)	0.28
Su vs. Pl	−1.0 (−1.5, −0.6)	−1.1 (−1.6, −0.7)	−0.3 (−1.7, 1.2)	0.28

Et: etanercept; In: infliximab; Mt: methotrexate; Pl: placebo; Su: sulfasalazine.

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
