# Peer review of "Different Original and Biosimilar TNF Inhibitors Similarly Reduce Joint Destruction in Rheumatoid Arthritis—A Network Meta-Analysis of 36 Randomized Controlled Trials"

_ijms, 2019, doi:10.3390/ijms20184350_

Round 1
Reviewer 1 Report
ijms-558241
Different reference and biosimilar TNF inhibitors similarly reduce joint destruction in rheumatoid arthritis. A network meta-analysis of 36 randomized controlled trials.
This article provides a meta-analysis of 36 randomized control trials focusing on the efficacy of 5 established TNF inhibitors (TNFi; including TNF antibodies and a TNF receptor blocker) and 3 biosimilar products on joint destruction in RA. The effects of low, standard, and high doses were assessed and compared to available MTX or placebo controls. The authors report that the analyzed TNFi significantly reduced joint destruction either with or without additional MTX treatment. Below-standard TNFi doses also had significantly improved effects when compared to MTX monotherapy. The authors conclude that even limited TNFi treatment may be effective in a variety of RA patients. Moreover, biosimilar products showed comparable effects as the respective reference drugs and may represent a way to reduce the economic burden of RA treatment.
In my opinion, the study is very well designed, carried out properly, and technically and statistically sound. The results are comprehensible and conclusive. In addition, the manuscript appears straightforward and clear. Therefore, only a few a few minor aspects have to be addressed.
1. Page 2, section 2.1: Did the authors apply specific/defined exclusion criteria to reduce the number of initially selected articles? 2. Page 3, lines 118-119: Please provide the reference numbers of the studies without reported DAS28. 3. Page 4, line 124: The abbreviation RP (for randomization procedure) is missing. 4. Page 4, lines 130-132: Please provide the reference numbers of the studies i) without information concerning GC intake and ii) for which modified estimates were used (i.e., differing from 2.5 mg). 5. Page 5, concerning the biosimilars SB2, CTP, and SB4: Please include the information to which reference drugs these products are similar. 6. Page 6, Table 2, column “unadjusted”: in part, values defining the range are missing. 7. Page 9, line 289: In the text, a 'Table 5' is mentioned, but only 4 Tables are provided. Please clarify. 8. Page 14, lines 511-512: the author BS is mentioned twice for “writing - review and editing”. Please correct. 9. Page 14, line 516: Please include the grant number for the Danske Regioners Medicin Pulje grant. 10. A few typos/inconsistencies in style should be corrected (i.e., page 5, lines 177-179; page 6, lines 199-200; page 7, lines 207-209; Table 3; page 8, lines 220-221: font color; page 8, lines 225-239: inconsistent setting of spaces, e.g., 4w vs. 2 w and -0.2% vs. ‑ 0.9%; page 11, line 389: at the end of the sentence, the punctuation mark is missing).
Author Response
Thank you very much for your thorough and helpful comments which we think have contributed to improve the quality of the manuscript
1. Page 2, section 2.1: Did the authors apply specific/defined exclusion criteria to reduce the number of initially selected articles? No, there were no specific exclusion criteria. All studies fulfilling the inclusion criteria were included. we have added the following sentence: Line 412 There were no exclusion criteria for studies fulfilling all inclusion criteria
2. Page 3, lines 118-119: Please provide the reference numbers of the studies without reported DAS28.
We have added the reference numbers for the two studies reporting DAS44 (17, 25) and the 6 studies not reporting DAS28 (14-16, 18-19, 23).
3. Page 4, line 124: The abbreviation RP (for randomization procedure) is missing.
We have added this abbreviation
4. Page 4, lines 130-132: Please provide the reference numbers of the studies i) without information concerning GC intake and ii) for which modified estimates were used (i.e., differing from 2.5 mg).
We have rewritten this section to , hopefully, be more precise
Glucocorticoid (GC) was not allowed in two studies (17,20). In the remaining studies the mean GC dose per patient varied from 0.4 mg to 4.8 mg (Table 1, GC column). One study directly reported the mean GC intake in the treatment groups (34). Three studies did not report the GC treatment interval, but did report the percentage of patients being treated with GC (21,40,43). An estimated dose of 5 mg of GC multiplied by the percentage of treated patients was imputed for these 3 studies. Six studies only reported the interval of 0-10 mg of GC (32,36,39,45,47,48). Assuming a median dose of 5 mg and 50% of the patients receiving GC, an estimated GC intake of 2.5 mg was imputed for these 6 studies. The remaining studies reported both the interval of intake and the percentage of patient taking steroids. The median GC dose of the interval multiplied by the percentage of treated patients was imputed for these studies.
5. Page 5, concerning the biosimilars SB2, CTP, and SB4: Please include the information to which reference drugs these products are similar.
SB2 (infliximab), CTP (infliximab) and SB4 (etanercept) is now included
Page 6, Table 2, column “unadjusted”: in part, values defining the range are missing
They were hidden behind the right column line. I have dragged the column line to the right and the values are now visible
7. Page 9, line 289: In the text, a 'Table 5' is mentioned, but only 4 Tables are provided. Please clarify
Sorry, we are referring to Table 4, this has now been changed.
8. Page 14, lines 511-512: the author BS is mentioned twice for “writing - review and editing”. Please correct.
One "BS" removed, thank you for noticing
9. Page 14, line 516: Please include the grant number for the Danske Regioners Medicin Pulje grant.
Grant no (EMN 2017 00901) included
10. A few typos/inconsistencies in style should be corrected (i.e., page 5, lines 177-179; page 6, lines 199-200; page 7, lines 207-209; Table 3; page 8, lines 220-221: font color; page 8, lines 225-239: inconsistent setting of spaces, e.g., 4w vs. 2 w and -0.2% vs. ‑ 0.9%; page 11, line 389: at the end of the sentence, the punctuation mark is missing).
Thank you, these errors are corrected
Reviewer 2 Report
The meta-analysis is clear and well written. Authors did not find any difference among anti-TNF treatments, including biosimilars. However, I think that not seeing any difference among treatments is anyway interesting. The overall message of the manuscript is not new, but the authors checked some aspects that have never been checked. The attempt to include the placebo effect in the analysis is interesting even if the success of the attempt to include placebo in the network was limited.
Minor points.
- I think that in some points of the Result Section a more detailed description of the data presented in the Tables may improve the readability of the manuscript;
- Authors assume that the placebo effect is equal to no effect. In the discussion they must specify that the assumption may be wrong, thus further potentially increasing the effectiveness of the treatments;
- Few mistakes are present (line 233 and 513-514).
Author Response
Thank you very much for these relevant comments, which we think have contributed to improve the quality of the manuscript
I think that in some points of the Result Section a more detailed description of the data presented in the Tables may improve the readability of the manuscript;
We have added text in the results section to further describe some of the results and we have added references in the text to the tables to make it easier to find the relevant results in the tables. The changes are visible in the results section.
Authors assume that the placebo effect is equal to no effect. In the discussion they must specify that the assumption may be wrong, thus further potentially increasing the effectiveness of the treatments;
We have added the following lines in the limitation section of the discussion
The premise that placebo has no effect may be wrong and in that case the true effects may be even bigger than the present reported effects versus placebo.
Few mistakes are present (line 233 and 513-514).
Line 233: "from" eliminated
Lines 513-14 eliminated